# CNN-Based Model for Landslide Susceptibility Assessment from Multispectral Data

**Diego Renza** [1,*], **Elsa Adriana Cárdenas** [1], **Estibaliz Martinez** [2] **and Serena Sarah Weber** [3]

1. Faculty of Engineering, Universidad Militar Nueva Granada, Bogotá 110111, Colombia
2. Computer Systems Architecture and Technology, Universidad Politecnica de Madrid, Campus de Montegancedo, Boadilla del Monte, 28660 Madrid, Spain
3. Faculty of Engineering and Architecture, Universidad Católica de Manizales, Manizales 170002, Colombia
* Correspondence: diego.renza@unimilitar.edu.co

**Abstract:** In this work, a new convolutional neural network architecture is proposed to evaluate the susceptibility to landslides. It is a supervised learning algorithm that has been trained from data whose labels have been obtained by applying a heuristic method that involves geological, geomorphological and land use information. The attributes contemplated the use of multispectral data and spectral indices, in addition to slope and DEM data. Although the cartographic unit in the proposed method is the pixel, the processing was performed at the patch level since it involved the use of spatial information around each pixel. Therefore, the proposed deep learning architecture is characterized by its simplicity and by applying both spatial and channel processing. The proposed method presents similar performance to state-of-the-art methods, achieving an F1 score higher than 88% on test data with low computational cost and pixel-level accuracy.

**Keywords:** landslide; susceptibility; deep learning; CNN; GIS





## 1. Introduction

Mass movements, understood as the downward movement of rocks, debris or earth due to the action of gravity, have caused significant social and economic costs for more than half a century in Colombia, especially during rainy periods, affecting numerous populations and infrastructure works [1]. In general, the impact of hydro-meteorological phenomena (such as mass movements) can result in human suffering, including death and damage to infrastructure. Thus, in Colombia, 88% of disasters are associated with the occurrence of this type of event, and about 14% of affected homes and 66% of deaths are associated with mass movements [2].

In this regard, in order to identify and mitigate the effects generated by the occurrence of these events, as well as to facilitate more efficient management of the territory, several types of methods have been proposed. The application of these methodologies is oriented to evaluate the susceptibility to landslides, considering that it depends on aspects such as the type of movement, the spatial scale, the available information and the experience of those who apply it.

Subsequently, these methods involve models based on a combination of geological, topographical and land cover conditions, based on inventories and knowledge, data-driven quantitative methods or physics-based models. There are also heuristic methods that require the intervention of an expert in the field or are supported by geological and geomorphological terrain information and GIS (Geographic Information System) tools. In turn, bi-variate, multivariate and artificial neural network-based statistical methods have also been proposed, as well as physical evaluation methods obtained from the modeling of slope failure processes [3].

In addition, tools such as remote sensing have become important mechanisms for landslide identification and characterization, especially in mountainous areas. Beyond the

traditional techniques of visual interpretation and digital analysis of aerial photographs and satellite images [4–6], there are supervised classification methods to differentiate hill-slope landslides from other landcover units [7], and the availability of high-resolution images (including fused images) has become important [8,9]. It is also possible to combine remote sensing-based interpretations with heuristic and statistical susceptibility models [10] and to evaluate the effects of image features with respect to maps produced from different methods [11].

For example, in [12], earth observation data were used to map landslide risk by compiling regional-scale landslide inventory maps using satellite photo-interpretation and calculating displacement rates quantified from satellite SAR (synthetic-aperture radar) interferometry. Similarly, some alternatives have included geomorphological inventories of seasonal and multi-temporal events, and they have used other technologies related to high-resolution digital elevation models (DEM) [13], as in the analysis of the influence of tectonics on the progressive erosion of landscapes and their classification from the combination of DEM-based geomorphic indices [14]. Meanwhile, other proposed methodologies include typical morphometric parameters derived from DEM using "window-based" morphometric indices as basic predictor variables for the landslide susceptibility assessment [15]. For example, sixteen morphometric factors derived from a DEM using ASTER (Advanced Spaceborne Thermal Spaceborne Emission and Reflection Radiometer) data and other geological and environmental predictors were recently used [16].

Regarding the methods for modeling the susceptibility to landslides on a statistical basis, a critical review of the methods proposed in the last 25 years was presented in [17]. According to this study, different categories of models related to geographic area, type of landslide, type of inventory and period covered, statistical model, evaluation method, or uncertainty evaluation were identified. The study also concluded that the most common statistical methods included logistic regression, neural networks, data overlay, index-based analysis, and the trend towards machine learning-based methods, which has been corroborated in recent years. In addition, complementary studies have addressed the evaluation and comparison of traditional methods such as support vector machines [18], random forest [19], or ensemble and hybrid algorithms [20–23].

Regarding the use of deep learning methods, recent work has compared different neural network architectures while proposing new ones [24,25]. For instance, the authors in [25] propose the use of Convolutional Neural Networks (CNN) of two layers for landslide susceptibility mapping using three data representation forms (i.e., 1D, 2D or 3D). Hence, the 1D representation is obtained from the landslide influencing attributes, and  this 1D grid cell is converted first to a 2D matrix and then to a 3D representation to fit each CNN architecture. Thus, there is no guarantee that the data have a spatial relationship that can be exploited for feature extraction by the cross-correlation operation performed by the network. The proposal was evaluated with data from Yanshan County (China), involved 16 landslide influencing factors, and the landslide inventory map had only 400 locations distributed for training and validation.

Subsequently, a paper on the application of a deep learning model for the prediction of landslide susceptibility was presented. Although this proposed model relates some hyperparameters and the tool used for its implementation, it does not describe fundamental characteristics such as the dimensions of the filters or the number of layers of the network, which are necessary to replicate the model proposed therein. The proposal was evaluated with data from th Muong Lay district (Vietnam) and involved nine conditioning factors from 217 landslide events and 217 non-landslide samples [26].

Another recently presented work proposes the use of a CNN for feature extraction and the use of a DNN (deep learning network) to classify pixels into high and low susceptibility groups [27]. However, as in the previous case, the model proposal does not provide the necessary details to obtain a complete view of the model or to replicate it. Furthermore, the dataset used to train and validate the network (222 landslides) could be too small to

train a deep learning model, as in the previous cases. This work used 16 triggering factors, and it was evaluated with data from the Isfahan province in Iran.

In [28], a CNN of three convolutional layers plus two fully connected layers and a classification layer was proposed for landslide susceptibility mapping. The input data used in this study correspond to 11 causative factors of 405 landslides from Wenchuan County (China), which were structured into 30 m resolution raster data. The dimension of the input data to CNN was $17 \times 17$ pixels. Other works have proposed the use of software solutions (Microsoft Power BI) to explore data, analyze root-cause, identify key influences, and identify and explain anomalies through the use of landslide data and artificial intelligence (natural language processing and CNN) [29].

The use of recurrent neural networks (RNN) has also been proposed in recent works. For this purpose, 17 landslide influencing factors were used, each one as a single-band image, where each pixel was converted to a sequential sample from the perspective of importance. Furthermore, all factor layers were stacked together. This proposal was evaluated with data from Yongxin County (China) with 364 historical landslide locations. The proposal tested four RNN-related models: RNN, long short term memory (LSTM), gated recurrent unit (GRU) and simple recurrent unit (SRU) [30].

Of the existing methods in the literature, three important aspects stand out. The first has to do with the lack of clarity in the hyperparameters of some CNN models proposed in the literature, which hinders the replication of the models or their direct application in other geographical areas. The second has to do with the diversity of thematic variables used, which were identified in 596 different input thematic variables and reclassified into 23 classes in [17]. The third aspect has to do with the fact that, in some cases, the number of landslides used to train a model was too small for the extent of the study area to be geomorphologically relevant [17].

Under the above context, this paper presents a new deep learning architecture for landslide detection. The proposed architecture is characterized by its simplicity and by applying both spatial and channel-wise processing since the input samples are taken in the neighborhood around each pixel. For the training and validation of the method, reference data were created from geological and geomorphological information and GIS data from a study area in Colombia, and a feature evaluation was performed to define the attributes to be used as inputs in the model.

## 2. Data Preparation

Since we intend to approach the identification of landslide susceptible areas as a supervised classification problem, it is necessary to have data and labels extracted from an inventory. Most of the studies available in the state-of-the-art have described or used a single inventory, usually constructed from a geomorphological, event, multitemporal or historical point of view. Geomorphological inventories where the authors use the physical features of the landscape to identify the landslides have been preferred by researchers [17].

Taking into account that the present study aims at mapping landslide susceptible zones from a multispectral image (of a specific date) using deep learning and that the size of the dataset is key for the training of this type of model, it is proposed to perform the labeling from the results of a heuristic geomorphological inventory, which will be based on the crossing of weighted thematic layers (variables), according to the density of instability phenomena.

From the MS pixels data and its susceptibility labels, the proposed model will iterate to learn the relationship between the data and the labels, i.e., MS information at the pixel level will be used to try to predict the level of landslide susceptibility.

### 2.1. Labeling Using a Heuristic Methodology

The label of each pixel can be determined by applying a heuristic geomorphological method, such as the one specified by the Colombian National Geological Service [31]. The application of this methodology is based on information related to terrain variation,

morphogenetic environments and the degree of moisture and infiltration in the area that can affect soil strength. In consequence, the lithological composition of the rock (geology), the denudation environment in which it is formed (geomorphology) and the type of land use given to that area (land cover) are used.

First, for the determination of susceptibility by geological factors, three weighted components are used (weathering resistance of rocks ($R$), differences in rocks in terms of strength and directionality of mechanical properties ($T$), and fracture density ($F_D$)), which measures regional structural discontinuities in rock masses. Accordingly, the susceptibility by geologic factor is given by Equation (1),

$$G = 0.25R + 0.25T + 0.5F_D. \tag{1}$$

Regarding the susceptibility component due to geomorphology, the geomorphological units generated by the Colombian Geological Service are considered, which are delimited in more detail using a DEM (12.5 m) and satellite images. This element is then qualified according to its morphogenesis (i.e., the causes and processes that shaped the landscape [31]). With $M_g$ being morphogenesis and $M_m$ being morphometry, the susceptibility factor for geomorphology ($G_m$) is given by Equation (2),

$$G_m = 0.4M_g + 0.6M_m. \tag{2}$$

Thirdly, it is necessary to take into account that, in general, the areas where more movements occur are directly related to the type of land use (e.g., bare or uncovered soils), as well as to areas of greater slope.

Finally, the total landslide susceptibility is calculated from the weighting of the three previous factors (i.e., the susceptibility obtained by geology ($G$), by geomorphology ($G_m$) and by landcover ($C$)). The weighting of these three components is shown in Equation (3).

$$S = 0.2G + 0.6G_m + 0.2C. \tag{3}$$

### 2.2. Feature Selection from MS Data

The thematic variables used in previous studies can be classified into five categories: morphological, geological, land cover, hydrological and other variables. The variables related to terrain morphology are usually obtained by processing terrain elevation data (e.g., DEM) and are the most commonly used variables since they have proven to be effective in predicting the spatial distribution of landslides [17].

From the geological point of view, one of the most common types of information is related to rock type. In the case of land cover and land use, this type of data can be complemented with results from the processing of optical satellite imagery. In addition, variables that measure physical characteristics related to, for example, vegetation cover have also been included in studies due to the ease of calculation for large areas using satellite images [32]. Other types of variables that have been used are related to geo-environmental information and other data. Distance to faults is one of them, but its disadvantage is that it is not information that is always available, whereas distance to rivers is mainly related to hydrologically controlled events [17].

According to [17], new methods based on remote sensing information captured by sensors such as Landsat or Sentinel-2 satellites are expected to contribute to the solution of the landslide susceptibility modeling problem. In fact, the idea in the current research is to evaluate the use of multispectral images as the main starting information for the landslide susceptibility model, given their wide availability at different scales that would allow feeding landslide early warning systems. Nevertheless, it is also necessary to evaluate the possibility of adding other variables beyond such multispectral data. In order to select the features that will be used in the current research, two alternatives can be considered.

The first is to perform a feature engineering process to transform the data into potentially more meaningful representations based on domain knowledge (i.e., to combine

two or more features to create new ones). This can be achieved by arithmetic operations of the numerical characteristics (spectral indices) that are applied to the different spectral bands of an image per pixel. These indices allow highlighting the characteristics of the area, such as vegetation, humidity, water, bare soil, etc., which could be important when estimating the susceptibility of an area to landslides. Therefore, the use of consolidated indices in the literature allows adding domain knowledge to the dataset. Some spectral indices for feature engineering are listed below, and the specific formula of each index for several sensors can be consulted in [33].

- Normalized Difference Vegetation Index (NDVI)
- Soil Adjusted Vegetation Index (SAVI)
- Normalized Difference Moisture Index (NDMI)
- Moisture Stress Index (MSI)
- Bare Soil Index (BSI)
- Soil Brightness Index (SBI)
- Alteration (ALT)
- Ferrous Silicates (SF)
- Normalized Burned Ratio Index (NBRI)
- Normalized Difference Water Index (NDWI)
- Normalized Difference Snow Index (NDSI)
- Normalized Difference Glacier Index (NDGI)

The second involves the use of complementary pixel-level data such as DEM (as suggested in [17]) and the corresponding rate of change in elevation for DEM since they constitute important inputs to obtain the variables that compose the geometric characteristics of the slope.

Finally, it is also important to evaluate the contribution that each of the input features can have to a classifier. To obtain this contribution, the decrease in the impurity of each feature can be averaged, where the impurity is the measure on which the choice of the (locally) optimal condition of a node in a decision tree is based.

## 3. Proposed Landslide Susceptibility Identification Model

The adoption of convolutional neural networks (CNN) in the context of the ImageNet Challenge is one of the keystones that has driven the progress of the computer vision field in recent years. Recently, in addition to CNN, other types of architectures have also been important tools for object recognition. For example, Transformers have been shown to work well not only for natural language processing applications but also for computer vision, proving to be an alternative for convolutional networks [34].

Similarly, recent research has led to proposals based on classical architectures that are competitive with those widely adopted. For example, the Brain Team at Google Research proposed the MLP-Mixer architecture, which does not use convolutions or self-attention, so it is based only on basic matrix multiplication routines [35]. This same team also proposed another MLP-based alternative to Transformers [34], which consists of channel projections, spatial projections, and gating, obtaining solid classification results in ImageNet [34]. These MLP-type architectures are competitive but conceptually and technically simpler alternatives to techniques such as Transformers or deep convolutional network architectures [34,35].

Indeed, the basis of these architectures is the multilayer perceptron (MLP), which basically consists of multiple layers of neurons, each fully connected to those in the layer below. When stacking many fully connected layers, usually the first L-1 layers are the representation, and the last layer is the linear predictor. This stacking also involves the use of a nonlinear activation function, which decides whether a neuron should be activated or not by calculating the weighted sum and adding the bias [36].

In view of the above, the architecture shown in Figure 1 is proposed for the identification of landslide susceptible zones. The proposed architecture uses two blocks consisting of

a convolutional layer, a Batch normalization (*BN*) layer, *GELU* (Gaussian error linear unit) activation function, and an additional convolution layer without an activation function.

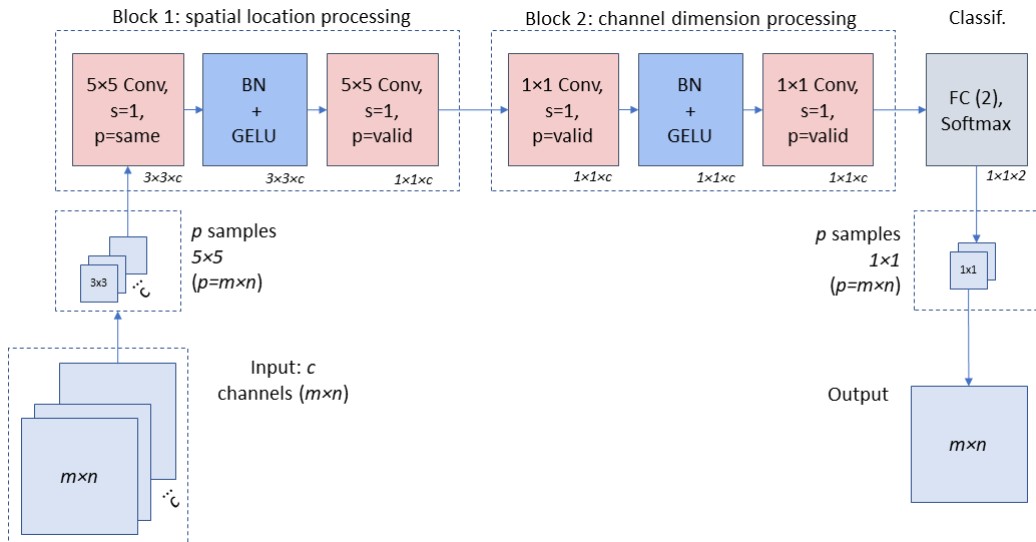

**Figure 1.** Proposed model for landslide susceptibility identification by means of remote sensing images and deep learning.

The first block uses a convolutional layer with $n$ filters, filter size $5 \times 5$, padding *same* and stride 1. The other convolutional layer uses the same number of filters and the same filter size, but a padding *valid*. The second block differs by using a $1 \times 1$ filter and padding *valid* in both convolutions. The idea in using the first block is to involve the spatial information surrounding the pixel by convolutions in a $5 \times 5$ neighborhood and, in the second block, to perform $1 \times 1$ convolutions for channel mixing.

As discussed so far, *BN* is used in each block after the first convolution. *BN* is a technique that systematically accelerates convergence in deep network training. It basically consists of normalizing an input by subtracting the mean and dividing by its standard deviation and then applying a scaling coefficient and a scaling offset, values that are learned during training [37].

Regarding the activation function, *GELU* is used as in the MLP-based proposals [34,35] after *BN*. For an input $x$ and a standard Gaussian cumulative distribution function $\phi(x)$, the *GELU* activation function is defined as $x\phi(x)$. The non-linearity of *GELU* weights the inputs by their value, unlike in *ReLU* where they are activated by their sign [38,39].

In the case of the input of the network, it is a tensor of size $5 \times 5$ extracted around each pixel of the input image, stacked in the channel dimension. The result is a 4D tensor indexed by batch, row, column, and depth. This means that for an image of $p$ pixels, the total number of input samples is the number of pixels, and the 4D tensor dimension will be $p \times 5 \times 5 \times c$ (where $c$ is the number of channels of the input image).

The selection of the input data dimensions is an important aspect since texture is a characteristic that can be related to landslide susceptibility, which, in turn, can be recognized by the CNN. In this sense, three sizes, namely $3 \times 3$, $5 \times 5$ and $7 \times 7$, were evaluated, taking into account the spatial resolution of the data (20 m), the depth of the proposed CNN and the downsampling (pooling) operations performed on the network. Although the best results were obtained with a $5 \times 5$ dimension, for data with different spatial resolution, or even for deeper networks, it is recommended to evaluate larger window sizes. One aspect to consider here has to do with the computational cost since the larger the window size, the higher the computational cost, due to the fact that for each pixel, a neighborhood with the established dimensions is extracted.

Having said that, the first block allows communication between different spatial locations (a $5 \times 5$ neighborhood); it operates simultaneously on all channels and its num-

ber of filters equals the number of channels of the input data. After the first convolution+*BN+GELU* the output size is $b \times 5 \times 5 \times c$ (where $b$ is the batch size). After the second convolution of this first block, the output size is $b \times 1 \times 1 \times c$, i.e., the data that had been taken in a $5 \times 5$ neighborhood are now processed at the pixel level.

For the second block, the computation occurs on the channel dimension, using $1 \times 1$ convolutions. In this case, it is important to note that a $1 \times 1$ convolutional layer is equivalent to an *FC* layer when applied on a per-pixel basis. After the second block is applied, the output size remains at $b \times 1 \times 1 \times c$.

Finally, a fully connected layer is applied at the end of the network to obtain the probability of each pixel belonging to one of the two classes. Considering the $p$ input examples (i.e., $p$ pixels of the input image), we will have output $p$ values related to the level of susceptibility to landslides at each position or geographical point represented by each pixel.

## 4. Implementation and Evaluation of the Method: A Case Study in Boyacá, Colombia

The procedure and results of the implementation and evaluation of the proposed method in a zone highly prone to landslides in Colombian territory are shown below. All source codes were implemented in Python using the TensorFlow framework (https://www.tensorflow.org, accessed on 30 June 2022) and the high-level API for TensorFlow, known as keras (https://keras.io, accessed on 30 June 2022). This was conducted using the Google CoLaboratory platform (https://research.google/tools/, accessed on 30 June 2022), which is a Jupyter notebook environment that does not require any configuration to use and runs entirely in the cloud. In addition to facilitating shared work, CoLaboratory also offers the possibility to access a GPU.

### 4.1. Study Area and Dataset

In order to evaluate the proposed methodology, an area with high susceptibility to mass movements was selected, given its geological, geomorphological and weather conditions. The zone is in the department of Boyacá, Colombia, and it includes rural areas in the municipalities of "San Luis de Gaceno", "Santa María" and "Chivor". The location of the study area is shown in Figure 2.

From the point of view of remote sensing data, the Sentinel-2A sensor was selected since it includes multiple bands in both the visible and infrared spectrum, with an acceptable spatial resolution (10, 20, and 60 m), as well as being available for free download and use. Accordingly, an MS image from the Sentinel-2A sensor captured in January 2016 was selected for the study area, where a $998 \times 750$ pixels crop and 12-bits radiometric resolution were used. Specifically, of the thirteen bands available in the Sentinel 2A image, those with a resolution of 10 or 20 m were selected, corresponding to bands 2-8A, 11 and 12. It should be noted that bands 1, 9 and 10 were not considered as input attributes due to their low spatial resolution (60 m). In the selected bands, for each pixel position, the input attributes correspond to the pixel value of each of the image bands. The false-color Sentinel 2A image of the study area is shown in Figure 3.

### 4.2. Features and Labels

The obtained susceptibility map by geology, geomorphology and land use using the heuristic method for the study area are shown in Figure 4a,b,c, respectively. The landslide susceptibility map obtained using the heuristic method for the study area is shown in Figure 4d.

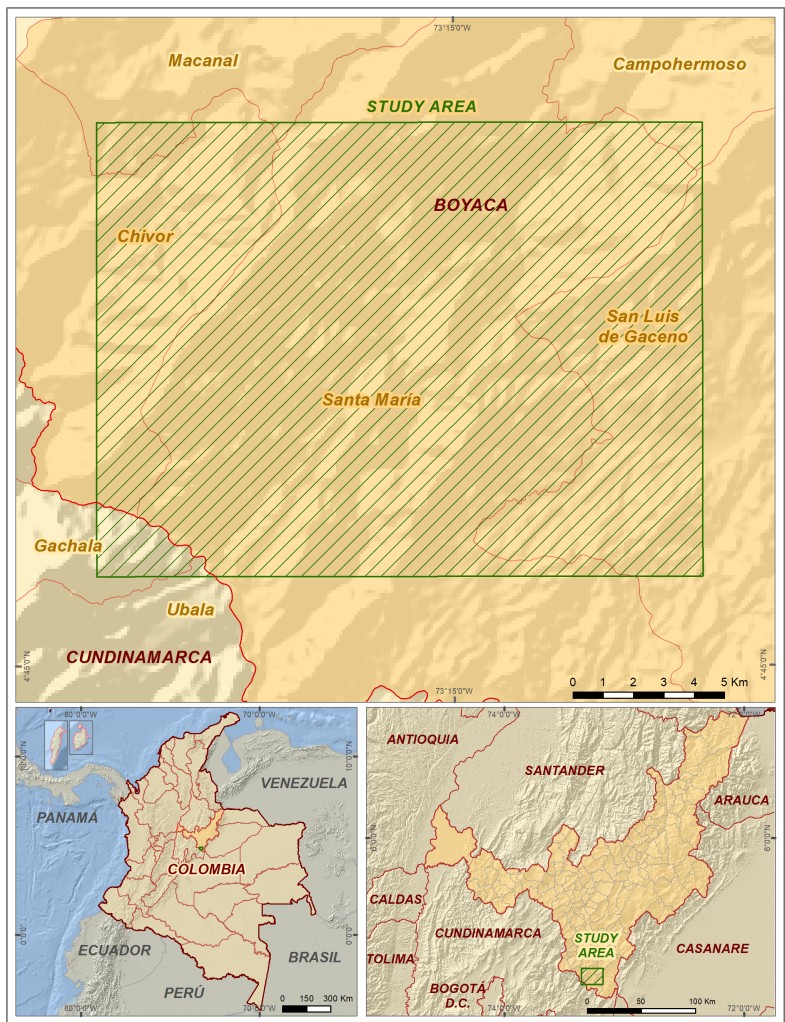

**Figure 2.** Location of the study area corresponding to the department of Boyacá, Colombia.

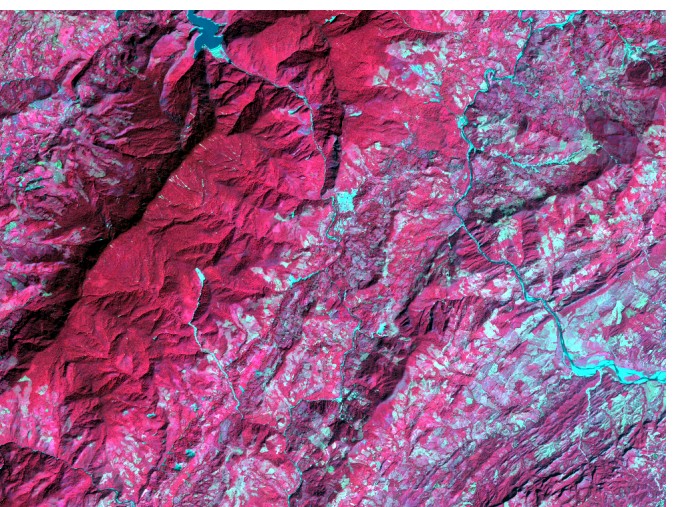

**Figure 3.** False color NIR, Red and Green composition of the study area using a Sentinel 2A image. The geographical zone covers an area of 300 km$^2$. The upper left corner of the zone is placed at 4°54′44.02″ N, −73°21′22.15″ W.

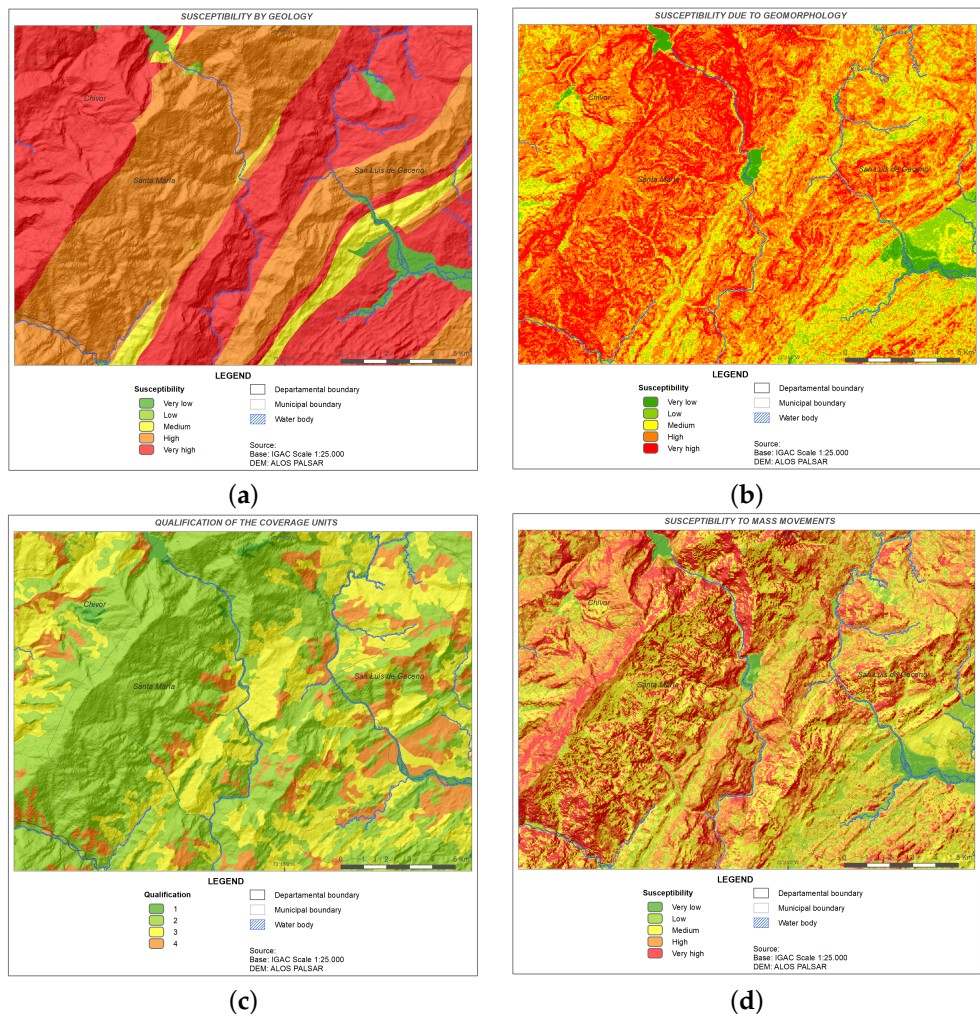

**Figure 4.** Inputs and final ground truth map for the study area. (**a**) Map of susceptibility by geology; (**b**) map of susceptibility by geomorphology; (**c**) coverage unit qualification map; (**d**) landslide movement susceptibility map.

It is important to note that the susceptibility of the map obtained using the heuristic method was defined on a scale of 1–5, with 5 being the highest level of susceptibility. Since the thematic variables selected for the present study involve only information derived from spectral data and DEM data, the performance of the algorithm is challenging in comparison with the results of a heuristic method that has involved morphological, geological and land cover variables with pixel-level accuracy. Taking into account that the scope of the proposed method is to obtain an initial estimate of the landslide susceptibility, it was chosen to reduce these five susceptibility levels to two classes: high susceptibility (for value 5) and moderate susceptibility (for values between 1 and 4).

Having said that, and in order to remove light regions that are smaller than a structuring element in the data, the morphological erosion operation was used. This operation converts a given pixel to the minimum pixel value in a neighborhood centered on that pixel. When applying the erosion operation, a flat disk-shaped structuring element with a radius of 1 was used to define the neighborhood (here, a pixel is within the neighborhood if the Euclidean distance between it and the origin is not greater than the radius) [40].

Regarding the feature selection, Figure 5 shows the result of the correlations between the susceptibility value obtained from the heuristic model and the pixel values in each MS band.

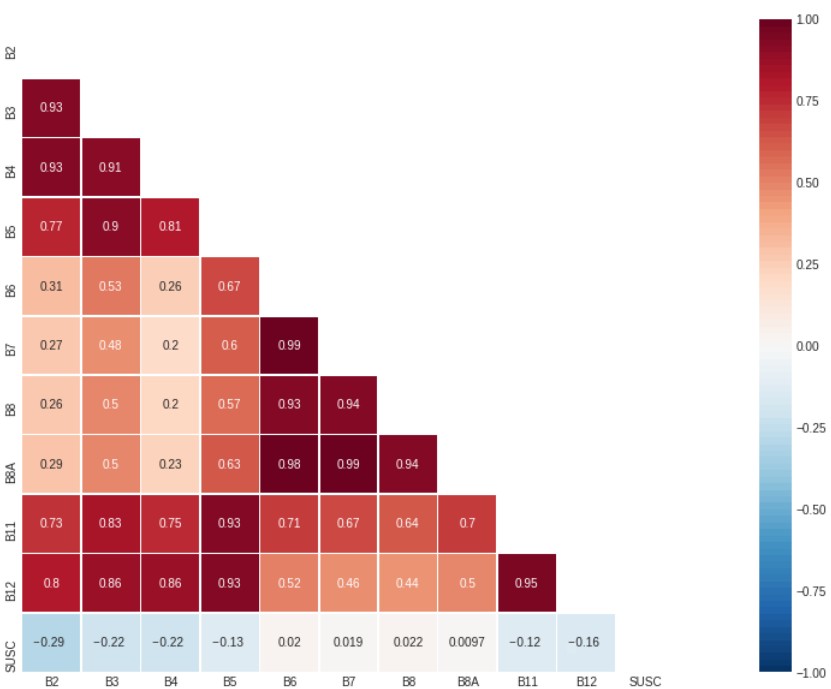

**Figure 5.** Correlations between bands (B2–B8, B8A, B11, B12) and the susceptibility label (SUSC).

As shown in Figure 5, the bands have a high degree of positive correlation with each other, while they tend to have a low and negative degree of correlation with the label. Indeed, only three bands (2, 3 and 4) have a correlation higher than 20% (in absolute value) with the susceptibility label. Following this, a feature engineering process was performed using the spectral indices listed in Section 2.2, a DEM was obtained from the Japanese ALOS PALSAR satellite (12.5 m of spatial resolution) and slope data were calculated using ArcGIS Desktop 10.7.

It is important to consider here that the original image bands (prior to index calculation) together with the DEM were normalized by dividing by 4095 ($2^{12} - 1$), while the slope was normalized by dividing by 90 (degrees). Likewise, DEM, slope and the indices were added to the multiband file, resampled to 20 m (nearest neighbor), maintaining the UTM WGS-84 projection. The correlation of the new set of features with each other and the correlation of each variable with the label are shown in Figure 6.

As depicted in Figure 6, some of the indices have a positive correlation with the output label (Susceptibility, SUSC); likewise, of the twelve indexes evaluated, eight show a correlation equal to or higher than 20% (in absolute value). On the other hand, the DEM has a positive correlation of 28%, while slope has the highest positive correlation (65%). Therefore, DEM and slope are necessary for model training, and some indices could serve as input features in a landslide susceptibility zone classification model.

Following this, the importance of each attribute was evaluated, obtaining the results shown in Figure 7.

As illustrated in Figure 7, the most contributing attribute in the classifier is slope, followed by DEM as expected from the correlation analysis. The bands and spectral indices contribute to a lesser extent, with a similar contribution. In this regard, the attributes whose correlation (in absolute value) is equal to or greater than 0.2 were selected, i.e., B2, B3, B4, NDVI, NDMI, MSI, BSI, ALT, SF, NBRI, NDGI, DEM and slope.

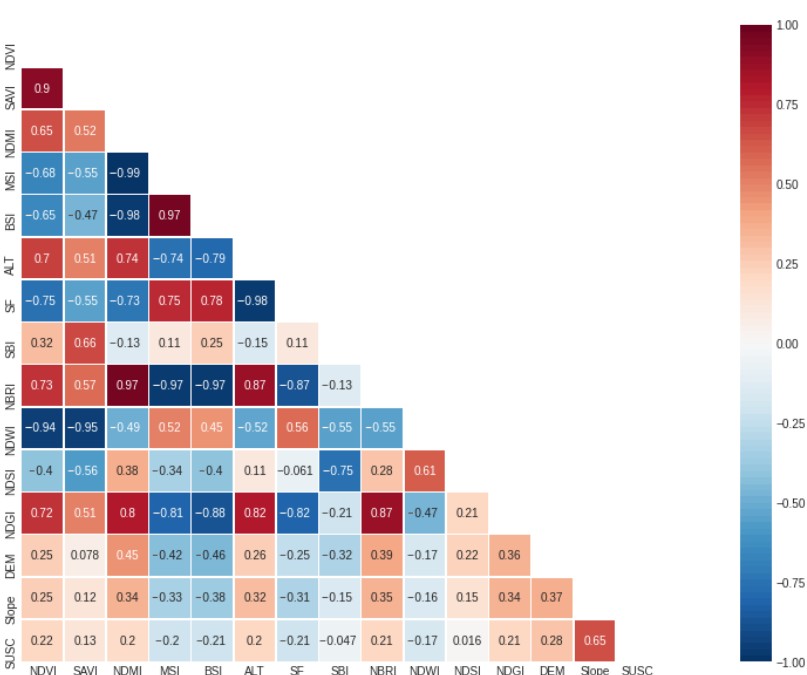

**Figure 6.** Correlations between bands (B2–B8, B8A, B11, B12), 12 spectral indices (see the list in Table 1), DEM, slope and the susceptibility label (SUSC).

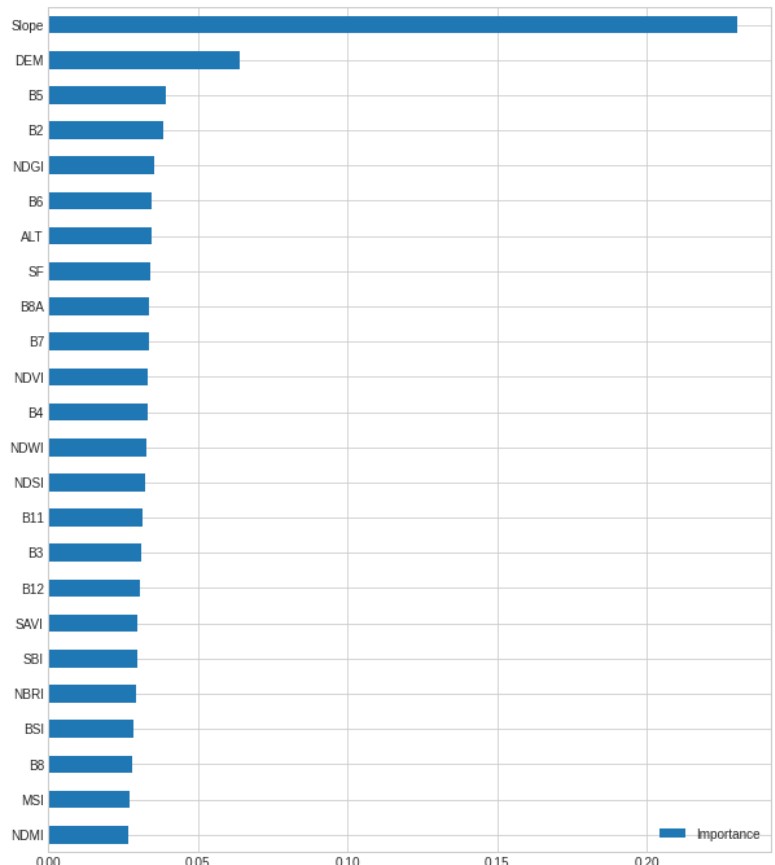

**Figure 7.** Feature importance calculated for 10 bands (B2–B8, B8A, B11, B12), 12 spectral indices, DEM and slope in a landslide susceptibility classification model.

**Table 1.** Evaluation metrics of the proposed model for the test set.

| Metric | Formula | Value (%) |
|---|---|---|
| Accuracy (*acc*) | $\frac{TP+TN}{TP+TN+FP+FN}$ | 87.73 |
| Precision (*P*) | $\frac{TP}{TP+FP}$ | 85.99 |
| Recall (*R*) | $\frac{TP}{TP+FN}$ | 90.35 |
| F1-Score | $2\frac{P\times R}{P+R}$ | 88.11 |

In conclusion of the above, the binary susceptibility value obtained using the heuristic method is used as the label for the training and evaluation of the proposed model, while the selected features include three bands from the original image, eight spectral indices, DEM and slope, and the problem is posed as a binary classification problem.

### 4.3. Model Training and Validation

As discussed above, the proposed architecture (Figure 1) was implemented and evaluated using Python, particularly the Keras/Tensorflow libraries, as well as the data described in Section 4.2. Furthermore, the Weights and Biases Dashboard [41] was used to adjust and select the hyperparameters of the model during training and validation. The results of the training, validation and testing of the model, as well as the comparison with the state-of-the-art, are shown below.

For the training and evaluation of the proposed model, 68,971 samples of high susceptibility and 68,971 samples of moderate susceptibility were selected, i.e., we worked with a balance between the two classes. These 137,942 samples were randomized, taking 80% of them as training data, 10% for hyperparameter selection (validation) and the other 10% was used to evaluate the model performance.

During model training and validation, model accuracy and error (through losses) were monitored, which allowed the hyperparameters of the model to be adjusted. In particular, the hyperparameters selected for model training and validation involved an Adam optimizer with a learning rate of 0.0001, a batch size of 128, and 60 epochs in total, as well as the architecture shown in Figure 1. The results in terms of accuracy and loss are shown in Figures 8 and 9.

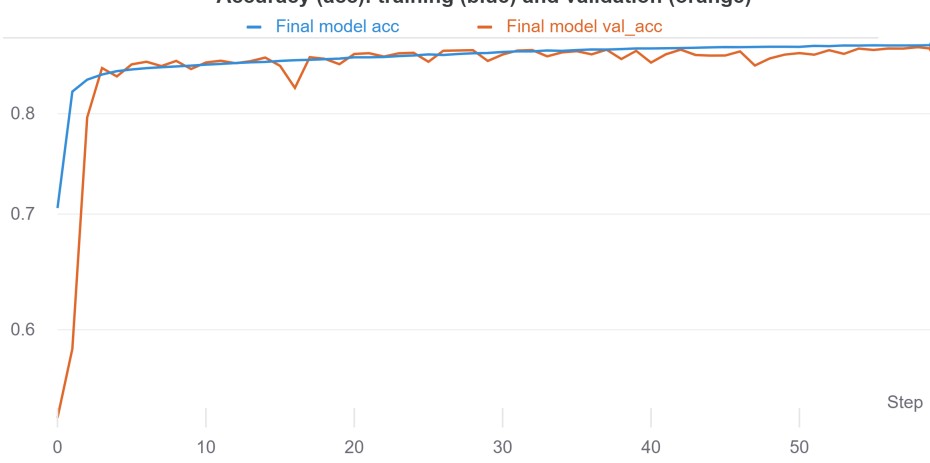

**Figure 8.** Plots of the percentage accuracy of the proposed model during training (blue) and validation (orange). Training accuracy reaches 88.31%, while the validation accuracy reaches 87.18%.

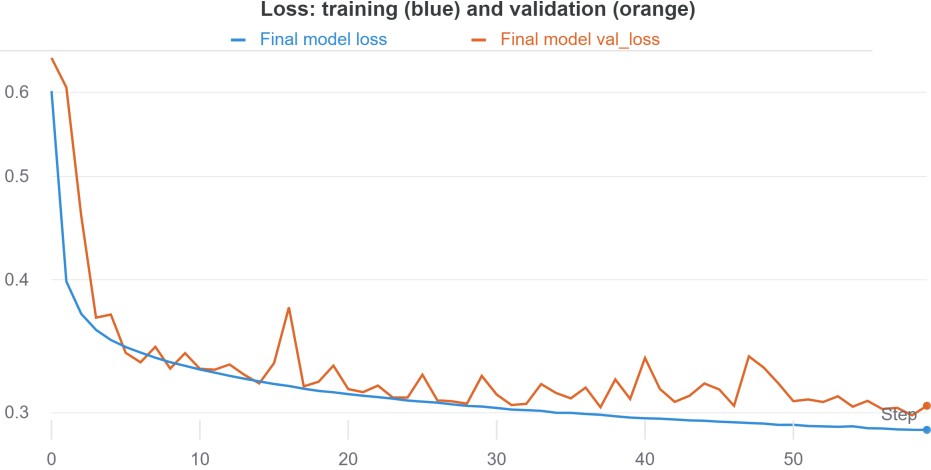

**Figure 9.** Plots of the loss during training (blue) and validation (orange) of the proposed model. After 50 epochs, the loss is reduced to 0.2749 and 0.2967 for training and validation, respectively.

According to Figures 8 and 9, little overfitting is observed since, in the validation, both accuracy and loss reach values similar to those of training. This is mainly due to the size of the dataset, as well as the use of regularization techniques such as *BN*. Figure 10 shows an example of the prediction of the trained model for the study area.

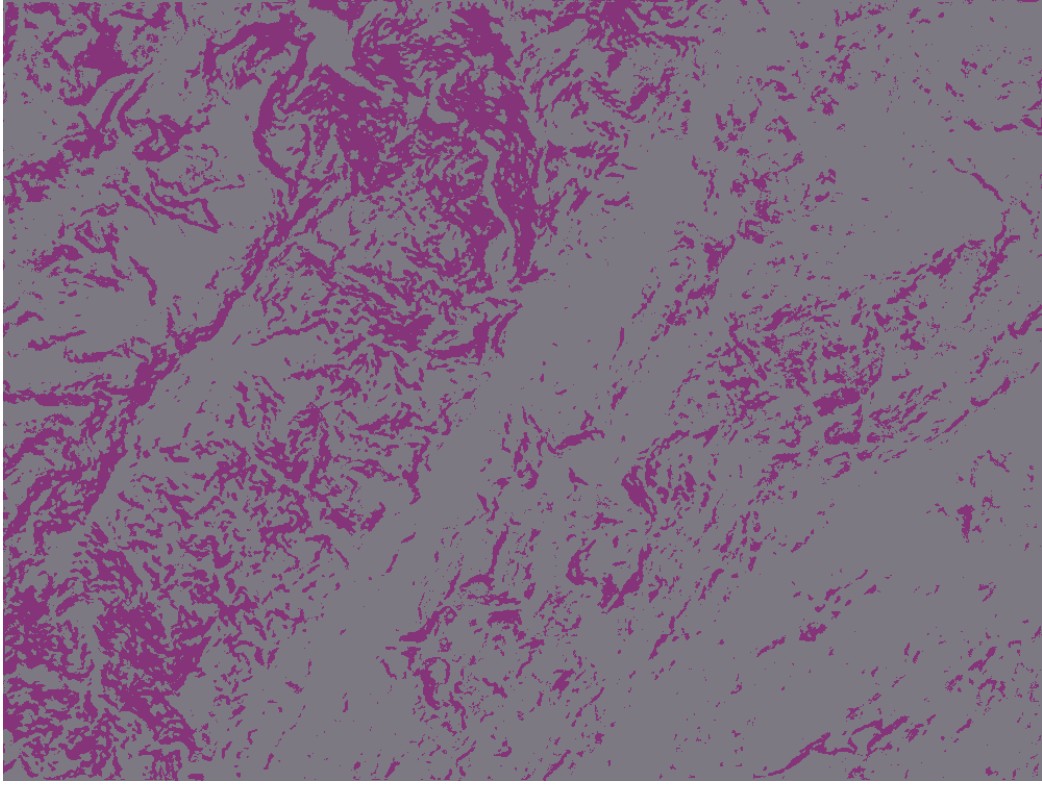

**Figure 10.** Example of a landslide susceptibility inventory using the proposed model for the study area. Purple: high susceptibility, Gray: moderate susceptibility.

### *4.4. Model Test*

To evaluate the performance of the proposed model, we used data unknown to the model, i.e., data that were used neither for training nor for validation. In this case, a total of 13,795 examples with the same distribution were used.

The results were consolidated into a binary confusion matrix, where *TP* (True Positives) and *TN* (True Negatives) correspond to correctly classified pixels (i.e., high susceptibility

and moderate susceptibility, respectively), and *FP* (False Negatives) and *FN* (False Positives) correspond to incorrectly classified pixels (i.e., moderate susceptibility classified as high susceptibility and high susceptibility classified as moderate susceptibility, respectively). The binary confusion matrix is shown in Figure 11, where $TN = 6272$, $FP = 1022$, $FN = 670$ and $TP = 5831$. From the values of the confusion matrix, the performance metrics shown in Table 1 were calculated. Firstly, the accuracy reached a value close to 88%, which means that most of the positive and negative samples were correctly classified.

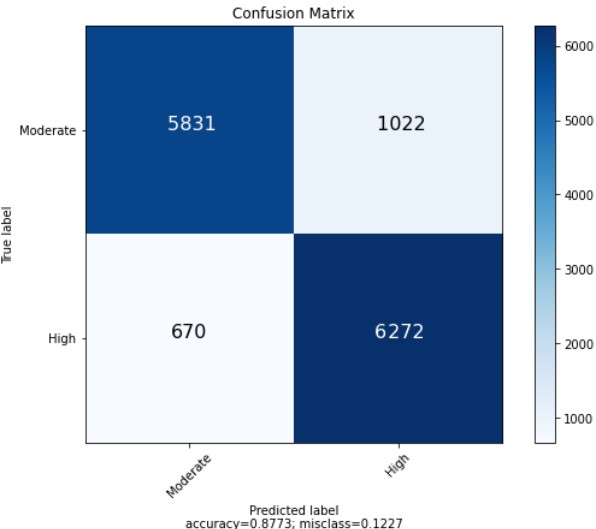

**Figure 11.** Confusion matrix for the test set (13,795 samples). The positive class corresponds to high susceptibility.

Regarding precision, a value similar to that of accuracy was achieved; however, this metric represents the percentage of classifications in the positive class that is actually identified as true positives. Sensitivity or recall, which represents the proportion of positives that are correctly identified, reached a slightly higher value (90%).

Moreover, the range of values that can be taken by the three metrics discussed so far is between 0 and 1 (0–100%), with the ideal value being 1. Since the difference in their formula differs only in the denominator (using *FP* or *FN*), when one of them increases, the other is likely to decrease. For this reason, it is important to assess the balance between them, for example, through an average. This can be obtained precisely by means of the F1-score, which corresponds to the harmonic mean between *P* and *R*. For the case evaluated, an F1-score of 88.11 was obtained.

## 5. Discussion

In order to compare the proposed algorithm with some state-of-the-art approaches, it must be taken into account that the number of susceptibility classes or levels in recent studies is not unified since different studies have been carried out that consider two, three or five levels, for example. In this regard, for comparison purposes, a recent study was selected that also worked with two levels of susceptibility and evaluated different machine learning techniques [27]. Therefore, some methods from this study were selected, including results from a proposal based on a deep convolutional neural network (see Table 2).

From the comparative study, the scheme based on Multi Layer Perceptron (MLP), the scheme based on Decision Tree and the proposed method in that paper were selected (based on deep convolutional neural network) [27]. Although that paper refers to the list of hyperparameters of the compared models, it does not clarify which values were finally selected. Even in the model proposed in that paper, hyperparameters such as the number and type of hidden layers or the number of units and filters are not given. It is

also important to note that this predictive model was applied in the evaluation of landslide susceptibility in the Isfahan Province, Iran.

**Table 2.** Comparison with state-of-the-art approaches. This table is included as a qualitative type comparison since the compared methods have been designed and trained under data sets with a significant difference in both the source and size of the datasets.

| Model | Accuracy | F1-Score | Dataset Size | No. of Parameters | Patch Size |
|---|---|---|---|---|---|
| MLP [17] | 61.0 | 38.0 | 222 samples | Not given | Not given |
| Decision Tree [17] | 80.0 | 80.0 | 222 samples | Not given | Not given |
| CNN-DNN [17] | 84.8 | 87.0 | 222 samples | Not given | Not given |
| Proposed | 87.73 | 88.11 | 137,942 samples | 8972 | $5 \times 5$ |

According to Table 2, the accuracy and the F1-score of the proposed method outperforms the score obtained by the machine learning and deep learning methods reported in [27]. Although the data used for the evaluation of the proposed method and the methods presented in [27] are different, something important to consider in this comparison has to do with the size of the test dataset used, as well as the balance of the classes in it. In particular, the use of metrics such as F1-score in unbalanced classes may bias the result, depending on the distribution of the classes. The results reported in [27] were calculated using a test dataset of 44 samples, whereas the size of the test dataset in the proposed method was 13,795 samples.

Regarding the labeling and input data size (image patch), it is important to consider whether the evaluation was performed at the pixel level or on a per-patch basis. Although the CNN-DNN method reported in [27] speaks of a patch analysis, the training data speak of landslide locations (i.e., points). In the proposed method, the input training data (features) are constructed from an MS image extracting patches with a size of $5 \times 5$ around each pixel. Therefore, the total number of examples corresponds to the same number of pixels in the image since the neighborhood of the pixel with a stride 1 is taken. Hence, the data are labeled at the pixel level. The importance of this aspect lies in the requirements imposed on the classifier since pixel-level labeling is more demanding than patch-level labeling (where the label is averaged over the entire patch).

From the network complexity point of view, the first piece of information is the number of convolutional layers in the network (see Table 2). Beyond this, it is important to consider other aspects, such as the size of the convolution filters or the number of units in fully connected layers. This allows calculating the total number of network parameters (trainable and non-trainable), which are associated with the size (MB) of the architecture. For the case of the proposed method, the network has only 8972 parameters, of which 8920 are trainable. This small number of parameters is mainly due to the small size of the input data and the depth of the network and is significantly smaller compared to the size of standard deep learning architectures that can have millions of parameters. Finally, it is also necessary to consider the number of input data channels used in the schemes (which is equivalent to the number of thematic variables in the proposed method) since this input influences the computational cost of the architecture.

From the geomorphological point of view, it is important to take into account that the area of analysis is located in a sector of the Eastern Cordillera of Colombia (belonging to the Andes Mountains), where high slopes dominate and where geoforms of structural origin have an areal distribution close to 90%. Therefore, it is considered that the factors that most influence landslide susceptibility are those of a geomorphological nature. Among these factors are the slope of the terrain, curvature, morphogenesis and other denudational processes that are indicators of instability.

Understanding that susceptibility is closely related to geomorphology, whose components are essential parameters of the dynamics of the earth's surface, the DEM, therefore, constitutes a fundamental input to obtaining the geometry of the slopes and detecting other changes in the terrain that may be associated with active morphodynamic processes.

Following this, the model obtained from the proposed methodology shows a high correlation with the susceptibility map by geomorphology, where the areas corresponding to high susceptibility in the map have a similar distribution to the resulting areas, indicating that for similar geomorphological conditions, the proposed method can be used as a first indicator of the susceptibility to landslides in a given area.

According to this discussion, the proposed method presents a performance similar to state-of-the-art methods, being validated at pixel level with balanced data and metrics according to such distribution. In addition, the proposed architecture is both conceptually and technically simpler than other architectures, with a low computational cost.

## 6. Conclusions

The proposed deep learning architecture can be an alternative for landslide detection using remote sensing data and slope information. Some features related to spectral indices were evaluated and included as input attributes to the model; however, the most important feature was the slope. One of the main characteristics of the proposed method involves the use of pixel-level data without ignoring the context information of the pixel (both spatially and spectrally). Thus, the proposed architecture performs both spatial and channel-wise processing. This facilitates larger training data sizes and avoids the dependence on image patches. In comparison with state-of-the-art methods, it was found that the proposed scheme achieves similar performance with a low computational cost, being validated with balanced data, which supports the use of typical metrics in the state-of-the-art.

Although the proposed method can serve as a quick alternative to obtain an initial estimate of the landslide susceptibility of an area from multispectral data and DEM, these results could be complemented with other types of methodologies involving other thematic variables (morphological, geological, land cover, hydrological, etc.), depending on the area to be evaluated and the data available for that area. Furthermore, the possible inclusion of additional thematic variables to the proposed model could facilitate the implementation of a classifier with a larger number of susceptibility levels.

**Author Contributions:** Conceptualization, E.A.C.; data curation, D.R.; formal analysis, D.R.; funding acquisition, E.A.C.; investigation, S.S.W.; methodology, D.R. and E.M.; project administration, E.A.C.; resources, S.S.W.; supervision, E.M.; visualization, S.S.W.; writing—original draft, D.R.; writing—review and editing, E.M. All authors have read and agreed to the published version of the manuscript.

**Funding:** This product is a result of the research project INV-ING-3190 of 2020, "Identificación de zonas asociadas a procesos de remoción en masa a través de análisis de percepción remota" funded by the "Universidad Militar Nueva Granada-Vicerrectoría de Investigaciones".

**Data Availability Statement:** Not applicable.

**Acknowledgments:** The authors thank Jaramillo. C.M. for their participation in methodological aspects and in the elaboration of the cartographic products.

**Conflicts of Interest:** The authors declare no conflict of interest.

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
