# Peer review of "CNN-Based Model for Landslide Susceptibility Assessment from Multispectral Data"

_applsci, doi:10.3390/app12178483_

Round 1

Reviewer 1 Report

CNN being one of most popular approach in machine learning, the author presented this interesting study in using CNN on Landslide susceptibility assessment. This paper could be further improved before acceptance and following are my observations:

1)      Background on using CNN specifically for landslide analysis has not been properly laid out within Introduction section. The author should very briefly explain what has been accomplished (as reported by existing literatures) so far on analyzing landslides with CNN. Most importantly, the author should briefly discuss how this study links (similarities or dissimilarities) with existing studies that used CNN for landslide analysis. In this regard, it is recommended to add the following references among others (for highlighting similarities and dissimilarities of approach, significance & outcome):

·         Sufi, F., "AI-Landslide: Software for acquiring hidden insights from global landslide data using Artificial Intelligence", Software Impacts, Vol. 10, No. 100177, 2021

·         Sikui Zhang, Lin Bai, Yuanwei L, Weile Li, Mingli Xie, Comparing Convolutional Neural Network and Machine Learning Models in Landslide Susceptibility Mapping: A Case Study in Wenchuan County, Frontiers in Environmental Science, Vol. 10, No 886841, 2022

2) Material and Method section should clearly highlight the tools and techniques used for this study for the sake of research reproducibility. What software, tools, libraries, APIs were used for conducting this study? A subsection with these details would be nice.

3) Can this approach be used in other areas of research (e.g., Tornado susceptibility assessment)? If yes, then what are pre-conditions or adjustments required for applying this approach (if there is any)?

4) What are the limitations for this study? The authors should clearly highlight the limitation of this approach along with future improvements (within conclusion section).

Author Response

Responses in the attached file

Reviewer 2 Report

The manuscript proposed a convolutional neural network architecture to evaluate the susceptibility to landslides from multispectral data. The topic is interesting. However, there are some major concerns of mine that need to be resolved before it recommends acceptance,

1.     The biggest concern of mine is that the susceptibility to landslides showed quite low correlations with spectral features (Fig. 4 and Fig.5), but high with the slope (maybe the referenced susceptibility to landslides is calculated by the slope with a proportion of 60% see in formula (3)). The author should explain the feasibility of multispectral data in the detection of susceptibility to landslides with convincible evidence.

2.     The texture is an important feature that is recognized by CNN, which could also be highly related to the susceptibility of landslides. The present manuscript, however, mentioned using labels with the size of 5â…¹5 only. The number and criterion about how to select the labels were not clarified clearly. Also, the exact forms of the selected labels in RS images were not shown in the paper.

3.     The texture of the areas which are susceptible to landslides might varies with the scale of local morphology. Therefore, the reason for the selection of labels with the size of 5â…¹5 should be explained. The manuscript also mentioned that a patch-level process was conducted. Unfortunately, there was hardly any sentence about the susceptibility of slides at the patch level in the discussion section. Please provide more information.

4.     A five-classes level of susceptibility to landslides was determined in the referenced map. Why the predicted map (Fig. 10) is only in the two-classes level?

5.     The comparative results of the proposed method and other methods were shown in table 3. However, the results were based on different experimental contexts. It would be more convincible to compare the methods in the same conditions.

Moreover, other issues emerged in scientific writing. For example, the structure of the manuscript is not clear. The contributions of the study to the international scientific communities are not addressed well. The clarification of the method section is partially confusing. It would be better to restructure the manuscript for a more clear form.

Author Response

Responses in the attached file

Round 2

Reviewer 1 Report

All the requested changes have been made by the Authors. I have no further comments to add. Now the paper looks comprehensive and interesting.

Reviewer 2 Report

I have no more comments for scientific issues and the manuscript can be recommended after a go-throguh technical editing check.